# A phenopushing platform to identify compounds that alleviate acute hypoxic stress by fast-tracking cellular adaptation

Li Li[1,3], Heinz Hammerlindl [1,3], Susan Q. Shen[1,2,3], Feng Bao[1], Sabrina Hammerlindl[1], Steven J. Altschuler [1] ✉ & Lani F. Wu [1] ✉

Severe acute hypoxic stress is a major contributor to the pathology of human diseases, including ischemic disorders. Current treatments focus on managing consequences of hypoxia, with few addressing cellular adaptation to low-oxygen environments. Here, we investigate whether accelerating hypoxia adaptation could provide a strategy to alleviate acute hypoxic stress. We develop a high-content phenotypic screening platform to identify compounds that fast-track adaptation to hypoxic stress. Our platform captures a high-dimensional phenotypic hypoxia response trajectory consisting of normoxic, acutely stressed, and chronically adapted cell states. Leveraging this trajectory, we identify compounds that phenotypically shift cells from the acutely stressed state towards the adapted state, revealing mTOR/PI3K or BET inhibition as strategies to induce this phenotypic shift. Importantly, our compound hits promote the survival of liver cells exposed to ischemia-like stress, and rescue cardiomyocytes from hypoxic stress. Our "phenopushing" platform offers a general, target-agnostic approach to identify compounds and targets that accelerate cellular adaptation, applicable across various stress conditions.

Oxygen is indispensable for multicellular organisms, serving crucial roles in aerobic respiration, metabolism, and development[1,2]. However, hypoxia (low oxygen availability) underlies numerous disease states, including ischemic diseases, respiratory diseases, anemia, and non-alcoholic fatty liver disease[3,4]. Despite its pivotal role in disease pathology, pharmacological options to alleviate hypoxic stress are limited.

Cells are known to have endogenous mechanisms of hypoxia adaptation, which can be activated to protect against hypoxia-related stress[5,6]. For example, pre-exposure of tissues to hypoxia can protect against subsequent stress from ischemia (lack of oxygen and nutrients)[7–10]. However, the process of hypoxia adaptation takes time, during which cellular damage may have already occurred[11–13]. Accelerating hypoxia adaptation (i.e., fast-tracking acutely stressed cells

toward an adapted state) could serve as a therapeutic strategy to alleviate hypoxic stress.

The cellular response to hypoxia involves diverse and incompletely understood biological pathways[4,5]. The most well-studied of these pathways involve hypoxia-inducible factors (HIFs), transcription factors that are regulated in an oxygen-dependent manner by the prolyl hydroxylase domain enzymes (PHDs)[14,15]. However, the human genome encodes over 200 oxygen-dependent enzymes, each with the potential to regulate various cellular responses to hypoxia[16]. Accumulating evidence highlights the importance of non-HIF mechanisms (e.g., pathways involving KDM5A, KDM6A, and ADO) in sensing and responding to hypoxia[17–21]. Given this complexity, we hypothesized that a target-agnostic and high-dimensional approach would be effective in searching for pharmacological perturbations that accelerate adaptation to hypoxia.

[1]Department of Pharmaceutical Chemistry, University of California San Francisco, San Francisco, CA, USA. [2]Department of Psychiatry and Behavioral Sciences, University of California San Francisco, San Francisco, CA, USA. [3]These authors contributed equally: Li Li, Heinz Hammerlindl, Susan Q. Shen. ✉e-mail: steven.altschuler@ucsf.edu; lani.wu@ucsf.edu

Here, we present a phenotypic profiling platform to discover compounds and targets that can fast-track cellular adaptation to hypoxic stress (Fig. 1a). We first defined a high-dimensional space to track the progression of hypoxia response, which captures distinct phenotypes that reflect acutely stressed and chronically adapted states. Using this high-dimensional space, we screened an annotated compound library to identify compounds that shift cells toward the chronic hypoxia phenotype. Analysis of compound hits highlighted inhibition of mTOR/PI3K or BET activities as strategies to achieve this shift. Moreover, we demonstrated that this shift confers functional protection against hypoxia-related stress in orthogonal cellular assays.

Overall, our platform provides a framework for drug discovery aimed at alleviating cellular stress by fast-tracking adaptation.

## Results

### Phenotypic characterization of cellular hypoxia response trajectories

We sought to establish a high-content image-based cellular screening platform to model and capture the temporal dynamics of cellular response and adaptation to hypoxia. First, we searched for a cell model that would be both amenable to screening and have similarity to human tissue. In comparing 1034 cancer cell lines with 53 healthy

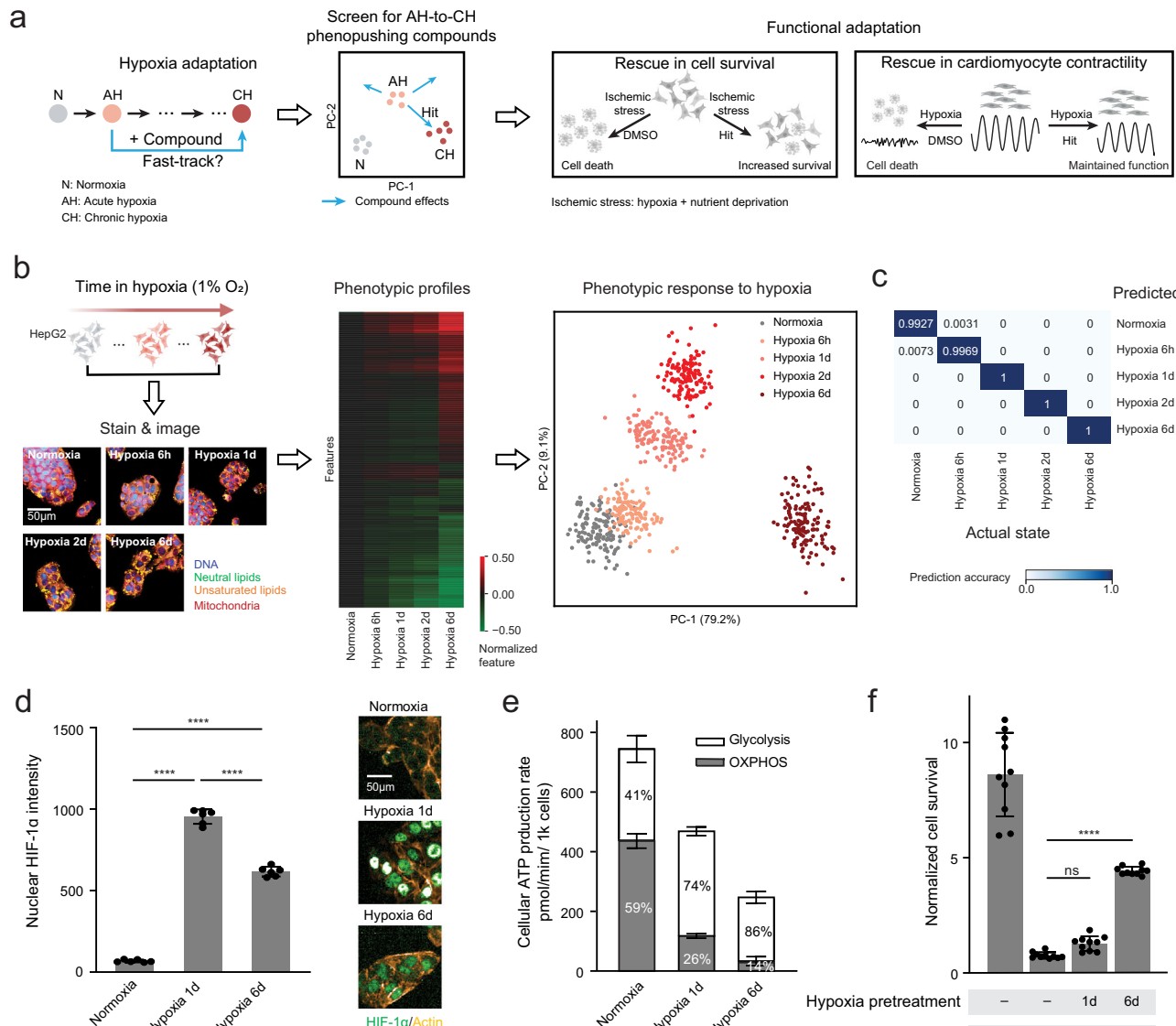

**Fig. 1 | Characterization of cellular states in response to hypoxia in HepG2 cells. a** Schematic overview of our study to identify and validate compounds that fast-track hypoxia adaptation. N: Normoxia. AH: Acute hypoxia. CH: Chronic hypoxia. The cell icons and beating pattern icons: Created in BioRender. Wu, A. (2025) https:// BioRender.com/q85j552. **b** Phenotypic profiling of cellular responses to hypoxia by metabolism-focused biomarkers at different timepoints (6 h, 1 d, 2 d and 6 d). Shown are representative results from 3 independent experiments with similar results. The cell icons: Created in BioRender. Wu, A. (2025) https://BioRender.com/q85j552. **c** Prediction accuracy for cellular states by kNN. **d** Nuclear HIF-1α levels in normoxia, hypoxia 1 d, and hypoxia 6 d. Left: mean nuclear HIF-1α per cell measured by immunofluorescence. Right: representative images of HIF-1α immunofluorescence.

Data are shown as mean ± SD from 6 biological replicates. One-way ANOVA followed by Tukey's post hoc test: ****: $p < 0.0001$. **e** Normalized ATP production rate from oxidative phosphorylation (OXPHOS) and glycolysis in normoxia, hypoxia 1 d, and hypoxia 6 d. Data are shown as mean ± SD from 6 biological replicates. Two-way ANOVA followed by Tukey's post hoc test: OXPHOS, $P < 0.0001$ for N vs Hypoxia 1 d, N vs Hypoxia 6 d, $p = 0.0018$ for Hypoxia 1 d vs Hypoxia 6 d; glycolysis, $p = 0.1137$ for N vs Hypoxia 1 d, $p = 0.0012$ for N vs Hypoxia 6 d, $p < 0.0001$ for Hypoxia 1 d vs Hypoxia 6 d. **f** Normalized cell survival under ischemic stress with different pre-treatment conditions. Data are shown as mean ± SD from 10 biological replicates. One-way ANOVA followed by Tukey's post hoc test: ns: $p = 0.6021$, ****$p < 0.0001$. Source data are provided as a Source Data file.

human tissues, we found that the liver cancer cell line, HepG2, and liver tissue exhibited the highest transcriptional similarity (Supplementary Fig. 1a, b)[22,23]. Given the liver's crucial role in hypoxia adaptation, HepG2 cells were chosen as the cell model for our study[24,25]. Second, we searched for experimental conditions to capture acute- and chronic-like hypoxic states. Guided by prior in vitro studies, we treated HepG2 cells with hypoxia (1% $O_2$) of various durations, ranging from six hours to six days (Fig. 1b)[26–29]. Third, we selected multiplexed image-based biomarkers to capture and distinguish these phenotypic states. Cells were labeled with biomarkers to capture expected hypoxia-induced changes to mitochondria (MitoTracker) and lipids (BODIPY 493/503 for neutral lipids and C11-BODIPY 581/591 for unsaturated lipids), as well as with a biomarker for segmenting cell images (Hoechst) (Fig. 1b). Finally, we summarized HepG2 cellular responses to hypoxia as high-dimensional phenotypic profiles (Methods, Supplementary Data 1)[30]. These profiles revealed a temporal-response trajectory to hypoxia (Fig. 1b). Multiple timepoints along the hypoxia trajectory could be distinguished phenotypically from each other and from normoxia with high accuracy (k-nearest neighbor with prediction accuracy >0.98; "Methods") (Fig. 1c).

Along the hypoxia trajectory, we examined whether the 1 d and 6 d timepoints could represent the acute and adapted hypoxic states (respectively). We observed significant accumulation of nuclear HIF-1α at 1 d and a lesser degree of accumulation by 6 d hypoxia (Fig. 1d), consistent with known activation and subsequent negative feedback in the HIF-PHD-VHL circuit[31–34]. Further, we observed a progressive shift from aerobic metabolism to glycolysis, accompanied by corresponding decreases in total ATP production (Fig. 1e)[5]. Finally, a key feature of hypoxia adaptation is tolerance to ischemia-like stress (hypoxia with deprivation of glucose and FBS)[8]. Importantly, we found that cells pretreated with 1 d of hypoxia were intolerant to ischemia-like stress. By contrast, cells pretreated with 6 d hypoxia showed tolerance to ischemia-like stress, suggesting that six days—but not one day—was sufficient to establish an adapted state (Fig. 1f). (We note that transcriptional profiles also captured distinct cellular hypoxic states, mirroring the image-based profiles; Supplementary Fig. 2a, b). Altogether, we identified a cellular system and high-dimensional phenotypic space in which normoxic (N), 1 d acute hypoxic (AH), and 6 d chronic adapted hypoxic (CH) states could be modeled and distinguished.

## Identification of compounds that phenopush cells from the acute hypoxic state towards the chronic hypoxic state

We next screened for perturbations that fast-track cellular adaptation to hypoxia, as measured by their ability to "phenopush" cells from the AH state towards the CH state. For our screening library, we curated 6011 well-characterized compounds (Supplementary Data 2) to facilitate target identification. Together, this library covered 1928 protein targets, representing ~50% of all druggable human proteins, and 98% (325) of all KEGG pathways (Fig. 2a, Supplementary Fig. 3)[35,36].

We investigated whether these compounds could alter cellular response to acute hypoxic stress. Cells were pretreated with compounds for 24 h and then exposed to acute hypoxia for 24 h for a total of 48 h compound treatment. At the end of the 48 h, cells were labeled, fixed, and imaged (Methods, Supplementary Fig. 4a). Drugs were screened at two concentrations and across multiple batches ("Methods"). To ensure that we captured consistent hypoxia and drug responses, we included three control plates in each batch: N control and CH control (the vehicle, DMSO) plates and a plate with reference compounds ("Methods", Supplementary Fig. 4a–e).

We summarized cellular responses as phenotypic profile vectors, calculated by comparing drug with DMSO treatment in the AH condition ("Methods"). We additionally computed N and CH phenotypic profiles through comparison with the AH condition ("Methods"). To establish phenotypic regions associated with the oxygen conditions, replicate DMSO wells were used to define N, AH, and CH "point

clouds". We then searched for "phenopushing hit" compounds whose profiles (at either tested concentration) are positioned outside of the AH cloud and towards the CH cloud (Fig. 2a). As a measurement of "phenopushing", we quantified: (1) bioactivity as measured by a phenotypic difference from the AH cloud, and (2) "phenopushing" as measured by the phenotypic shift of a perturbation in distance (D) and direction (θ) relative to the CH cloud (Methods, Fig. 2a). Bioactive compounds that met either the distance cutoff (Z Score ≤ −3) and/or direction cutoff (top 90% percentile of shift angles) were identified as phenopushing hits ("Methods", Fig. 2b).

In total, we identified 198 AH-to-CH phenopushing hit compounds ("Primary hits", Fig. 2c, Supplementary Data 3). A retest of these compounds over a wide range of doses confirmed that 92 (46%) passed the original hit-calling criteria ("Confirmed hits", Fig. 2c, Supplementary Data 3). Reassuringly, these confirmed hit compounds aligned with the chronic hypoxic state, as reflected by both the high-dimensional phenotypic profiles (Supplementary Fig. 4f) and images (Fig. 2d). Together, our strategy provided a list of high-confidence compounds that phenotypically push cells in an acute hypoxic state towards an adapted, chronic hypoxic state.

## Identification of mTOR and BET proteins as potential targets for phenopushing towards CH

We wondered whether the AH-to-CH phenopushing hits would reveal molecular targets related to hypoxia adaptation. Using the ChEMBL bioactivity database, we constructed a target activity profile for each screened compound (compound-to-target activity map, "Methods")[37]. We identified mTOR, PI3Ks, and BET family members (BRD2/3/4) as top targets overrepresented by hits compared to non-hits (Methods; Fig. 3a). In concordance with the overrepresentation analysis, compounds with higher potency for mTOR and BET proteins (and PI3Ks to some extent) were more likely to phenopush cells towards the adapted state compared to those with lower potency (Fig. 3b–d). Reassuringly, the potent phenopushing doses of these compounds were similar to potent doses reported in published studies for on-target cellular effects[38–41].

We next investigated whether downregulation of mTOR, PI3K, and/or BET activity is seen in the endogenous CH state, since pharmacologic inhibition of mTOR, PI3Ks, and BET proteins phenotypically mimicked the chronic state at an early timepoint. We monitored phosphorylation of S6 (S235/S236) and AKT (S473) as markers of mTORC1 and mTORC2 activity, respectively; phosphorylation of AKT (T308) as a marker of PI3Ks activity; and phosphorylation of RNA Pol II (S2) as a marker of BET protein activity[42–45]. We observed a progressive decrease in marker intensity for mTOR and BET proteins, but not PI3Ks, from normoxia to AH and then to CH (Fig. 3e–g, Supplementary Fig. 5a). Thus, downregulation of the activity of mTOR and BET proteins, but not of PI3Ks, is characteristic of the CH state. Consistent with the relevance of mTOR and BET proteins to the CH state, high hit rates in the screen were observed for mTOR-selective compounds (86%, 12/14), PI3K/mTOR dual-selective compounds (50%, 4/8), and BET-selective compounds (75%, 6/8), but not PI3K-selective compounds (14%, 3/22) (Supplementary Fig. 5b–g, Supplementary Data 4). Since PI3K and mTOR pathways are tightly coupled, and many mTOR-targeting compounds also target PI3Ks, we grouped compounds that target mTOR and/or PI3Ks into one hit compound category[44,46].

We wondered whether downregulation of mTOR and/or BET activity is a general feature of the AH-to-CH phenopushing hits. We assessed how all hits, at their phenopushing doses, affect mTOR and BET activities. First, we confirmed that in AH, relative to DMSO, all 36 mTOR/PI3K-targeting hits (regardless of selectivity) downregulated both pS6 (S235/S236) and pAkt (S473), reflecting inhibition of both mTORC1 and mTORC2 (Fig. 3h). Moreover, 73% (41/56) of other hits (including BET inhibitors) induced decreases in both pS6 (S235/S236) and pAkt (S473) (Fig. 3h). Similarly, all 5 BET-targeting hits and 91% (79/87) of other hits (including mTOR/PI3K inhibitors) induced a decrease

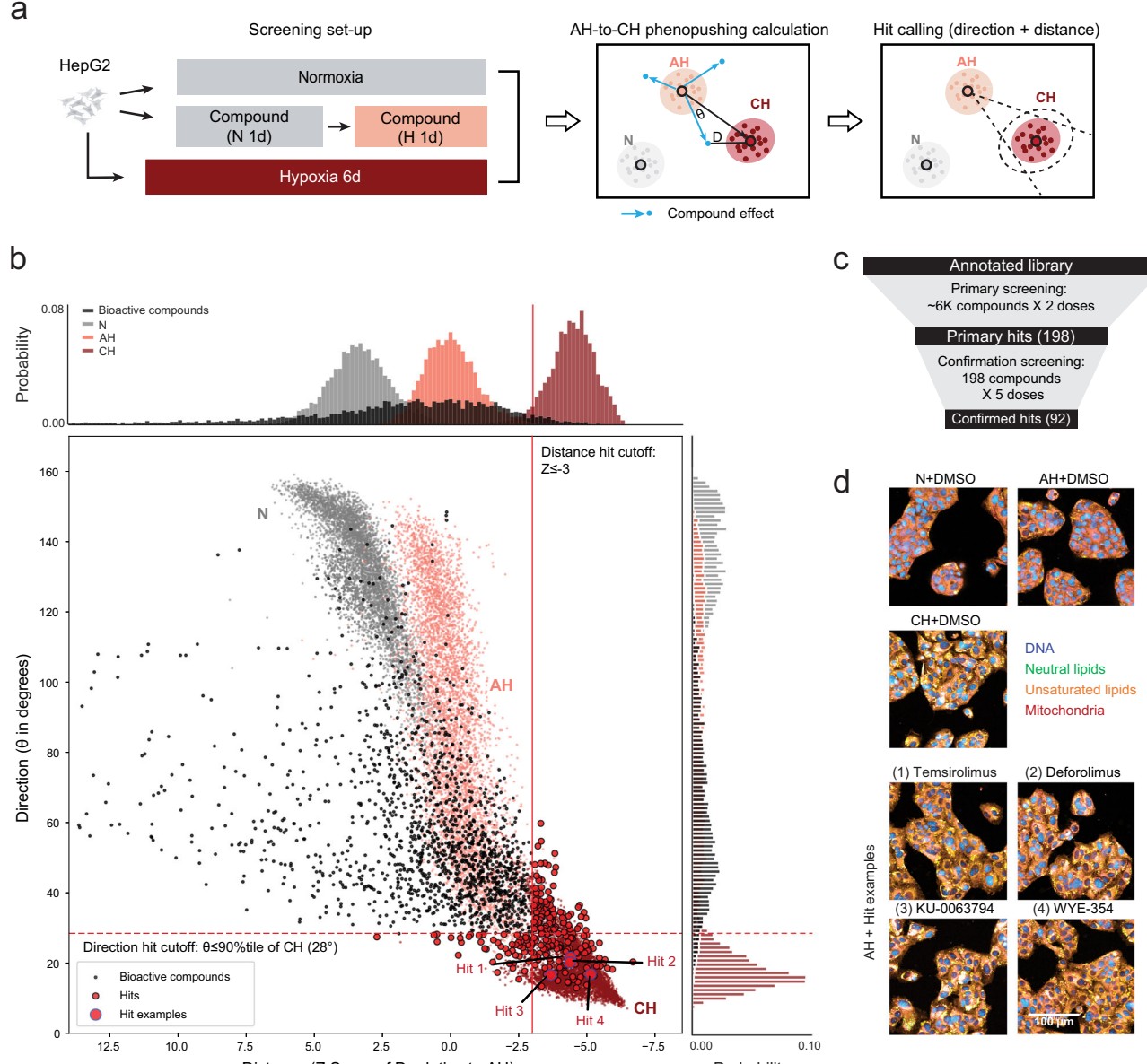

**Fig. 2 | Identification of compound hits that phenopush cells from the acute hypoxic state towards the chronic hypoxic state. a** Overview of AH-to-CH phenopushing screening and hit-calling framework. N: Normoxia. AH: Acute hypoxia (1 d). CH: Chronic hypoxia (6 d). D: Distance from the phenotypic profile of a perturbation (a given compound at a given dose) to the centroid of the phenotypic profiles of CH DMSO controls. θ: Angle of deviation from the phenotypic profile of a perturbation relative to the centroid of the phenotypic profiles of CH DMSO controls. The cell icons: Created in BioRender. Wu, A. (2025) https://BioRender.com/q85j552. **b** Summary of geometric hit calls in the primary screen.

Scatter plot and density map for distance (x-axis) and direction (y-axis) of bioactive compounds. Red solid line and dotted line show cutoffs for distance and angle, respectively. Vehicle (DMSO) controls in N, AH, and CH are shown for reference. Bioactive non-hit compounds (black dots), hits (red dots), and four hit examples (large red dots) are shown. **c** Screening funnel for AH-to-CH phenopushing hits. **d** Representative images for DMSO under N, AH, or CH compared to hit examples (highlighted in b) in AH (Temsirolimus, Deforolimus, KU-0063794, WYE-354). The experiment was independently repeated three times with similar results. Source data are provided as a Source Data file.

in pPol-II level (Fig. 3h). In fact, 79% (73/92) of all confirmed hits inhibited both mTOR and BET pathways compared with DMSO-treated AH cells. Together, we observed that the activities of mTOR/PI3K/BET monotonically decrease from N, to AH, to CH. The majority of hit compounds, applied at AH, accelerate this observed decrease towards lower CH levels.

### Hit compounds rescue HepG2 survival in a cellular model of ischemia

The hit compounds fast-track the development of cellular phenotypic features seen in the CH cellular state. However, it is unclear whether

this translates to fast-tracking functional protection of the cells. We next investigated whether phenopushing compounds fast-track functional protection against ischemia-like stress, motivated by the observation that HepG2 cells in the hypoxia-adapted CH state−but not the acutely stressed AH state−are tolerant to the combination of oxygen and nutrient (glucose/FBS) deprivation (Figs. 1f and 4a).

To determine if AH-to-CH phenopushing hit compounds increase tolerance to ischemia-like conditions, we compared survival of drug-treated to control (DMSO-treated) cells. We ranked hit compounds based on their average survival across the tested drug doses (Fig. 4b; ischemia exposure was calibrated so that ~10% of control cells survive

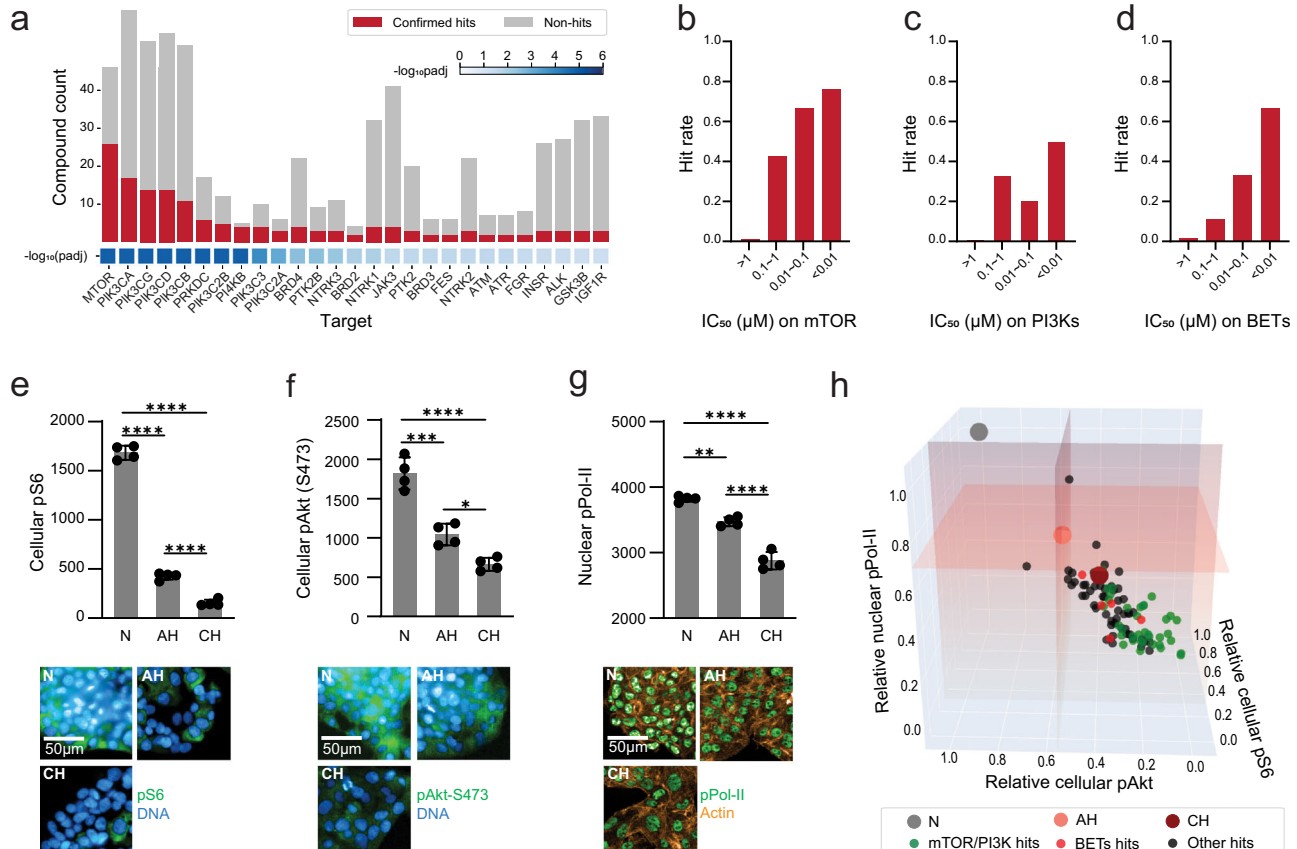

**Fig. 3 | mTOR/PI3K and BET proteins are targets for AH-to-CH phenopushing.**
**a** Enriched targets (by gene name) from overrepresentation analysis of hit compounds. Adjusted $p$ values are calculated by hypergeometric distribution overrepresentation analysis followed by Benjamini–Hochberg procedure correction. Bars: hit (red) or non-hit (gray) compound counts per target; Blue squares: adjusted p-values for overrepresentation analysis. **b–d** Hit rates of screened compounds annotated as targeting **b** mTOR, **c** PI3Ks and **d** BET proteins in different potency ranges (represented by $IC_{50}$). For PI3Ks and BET proteins, the median $IC_{50}$ across isoforms was used. **e** Cellular pS6 (at S235/S236), **f** cellular pAKT (at S473) and **g** Nuclear pPol II (at S2) in normoxia (N), acute hypoxia (AH, 1 d) and chronic hypoxia (CH, 6 d). Quantification of immunofluorescence intensity (well-level average of per-cell mean intensity) and representative images are shown for **e** cellular pS6 (at S235/S236), **f** cellular pAKT (at S473), or **g** nuclear pPol II (at S2). Data are shown as mean ± SD from 4 biological replicates for **e–g**. One-way ANOVA followed by Tukey's post hoc test: * in **f**: $p = 0.0149$, ** in **g**: $p = 0.001$, *** in **f**: $p = 0.0001$, **** in **e–g**: $p < 0.0001$. **h** Visualization of the effects of confirmed hits in AH on cellular pS6 (at S235/S236) (x-axis), cellular pAKT (at S473) (y-axis) and nuclear pPol II (at S2) (z-axis) based on immunofluorescence intensity (normalized to N DMSO controls). For statistical analyses, see Supplementary Data 4. Source data are provided as a Source Data file.

by the end of treatment). Our analysis showed that ~83% (75/90) of the tested phenopushing hits significantly outperformed the controls for survival in at least one of the tested doses, with the best compounds reaching survival close to that of the hypoxia-adapted CH state. Compounds associated with overrepresented targets (mTOR and/or PI3K, or BET) had the strongest survival benefit (Fig. 4b). The majority of the top 25 compounds increased survival compared to DMSO across all doses, and nearly all were associated with overrepresented targets (Fig. 4c).

We wondered whether our phenopushing approach identifies compounds enriched for protecting cells in ischemia-like stress. We measured cell survival in ischemia-like conditions for 102 compounds that were bioactive (i.e., pushed cells away from AH) but non-phenopushing (i.e., did not push towards CH). Re-analysis of cell survival for all compounds at the most protective tested dose showed that the AH-to-CH phenopushing hits were more likely to improve cell survival. For example, 41% of our phenopushing hits (Fig. 4d, red) vs. 3% of non-phenopushing bioactive compounds (Fig. 4d, blue) increased survival >3-fold compared to DMSO (Fig. 4d, red dashed line). Together, these results demonstrate that the AH-to-CH phenopushing hits also fast-track the development of ischemia-like tolerance, a hallmark of hypoxia adapted cells (Fig. 1f).

## Phenopushing compounds protect matured iPSC-derived cardiomyocytes against hypoxic stress

We next examined whether phenopushing compounds provide functional rescue in other cellular contexts beyond the originally screened HepG2 cell line. While HepG2 cells are well-suited to studying adaptation to hypoxia, matured iPSC-derived cardiomyocytes (iPSC-CMs) are well-suited to studying susceptibility to hypoxic stress[47–49]. Matured iPSC-CMs are highly sensitive to hypoxia and stop spontaneous beating within 48–72 h of exposure to 1% $O_2$. Further, matured iPSC-CMs provide a functional cell-health phenotype, namely spontaneous regular beating patterns (Fig. 5a).

We tested a subset of our phenopushing hits, focusing on mTOR/PI3K and BET inhibitors, for their ability to rescue cardiomyocyte contractility in acute hypoxia. As a proxy for cardiomyocyte beating, we monitored the $Ca^{2+}$ transients by live cell imaging the GFP fluorescence intensity of the calcium sensor GCaMP6, which was constitutively expressed in our iPSC-CMs. Principle components analysis (PCA) of features extracted from the resulting beating time series data revealed two clusters in which cardiomyocytes retained an ability to beat, with cluster #2 most similar to matured iPSC-CMs in normoxia (Fig. 5b). To complement cardiomyocyte beating, we further investigated sarcomeric structure of all hits in cluster 2 (Fig. 5c, d). As in the

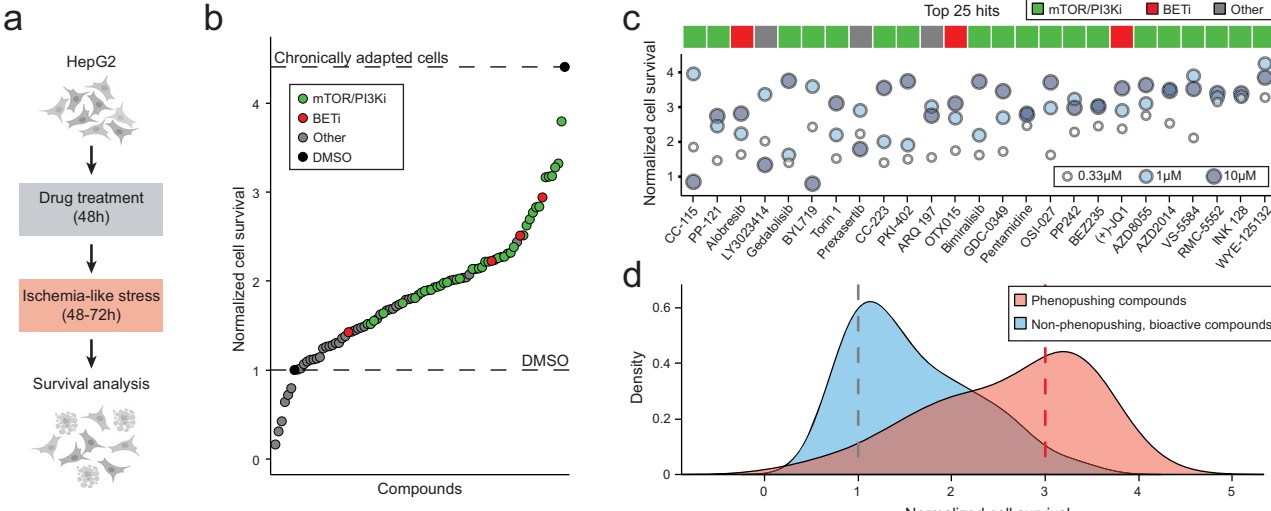

**Fig. 4 | AH-to-CH phenopushing hits rescue cells from ischemia-like stress.**
**a** Overview of the ischemia rescue assay. The cell icons: Created in BioRender. Wu, A. (2025) https://BioRender.com/q85j552. **b** Scatter plot of hit compound effects on HepG2 survival in ischemia-like stress. Cell survival: normalized to DMSO-treated cells, averaged across all tested well replicates and doses for each drug, and ranked based on their survival rate. Dashed lines: survival of DMSO-treated hypoxia naïve cells (bottom) or DMSO-treated cells pre-exposed to hypoxia for 6 days (top). mTOR/PI3Ki: AH-to-CH phenopushing hits targeting mTOR and/or PI3Ks. BETi: AH-to-CH phenopushing hits targeting BET proteins. Other: AH-to-CH phenopushing hits other than mTOR/PI3Ki and BET proteins. **c** Normalized cell survival of each tested dose for the top 25 hits shown in **b**. **d** Ischemia rescue effect comparison between the most effective tested dose of phenopushing (red) and non-pheno-pushing, bioactive (blue) compounds. Kernel density estimation used to smooth histograms. Dashed lines: (grey) mean survival of DMSO-treated cells or (red) 3x DMSO survival. Source data are provided as a Source Data file.

HepG2 ischemia assay, compounds associated with mTOR/PI3K and BET proteins were the most effective at rescuing iPSC-CMs from hypoxic stress, with cardiomyocyte beating often deteriorating before observable loss of sarcomere structure (Fig. 5c). These findings demonstrate that phenopushing in high-dimensional space can translate into functional protection across cell models.

## Discussion

Adaptation to hypoxia has long been recognized as a powerful mechanism to alleviate hypoxic or ischemic stress[8,10]. In this study, we developed an early-stage drug discovery platform to identify compounds that accelerate adaptation and confer functional protection against severe acute hypoxic stress. Using our platform, we captured functionally defined acute hypoxic (AH) versus chronic hypoxic (CH) states via high-content microscopy (Fig. 1), identified compounds and targets whose modulation phenopush cells from AH to CH states (Figs. 2 and 3), and validated that these hit compounds functionally rescue cells under severe hypoxia or ischemia-like conditions (Figs. 4 and 5).

Our approach leverages the ability to identify acute and chronically adapted hypoxia regions in a high-dimensional phenotypic space and to analyze compound effects relative to these regions. Traditional life/death screens poorly capture the complexity of hypoxia adaptation, as preventing cell death does not equate to inducing adaptation[50–52]. Alternatively, omics approaches (e.g., transcriptomics, proteomics, metabolomics) can capture high-dimensional readouts of molecular changes; however, their high cost and limited throughput hinder their use for large-scale screening[50–52]. In contrast, high-content phenotypic profiling, capable of capturing the complex phenotypes of cells as they transition from a hypoxia-stressed state to a hypoxia-adapted state, offers a scalable and efficient method to fulfill both requirements.

The concept of phenopushing can be applied to various choices of start or destination timepoints or conditions, as long as they are phenotypically separable. We chose the 1-day AH and 6-day CH timepoints out of practical considerations for

performing the screen. However, in principle, the screen could be performed at timepoints as early as 2 hours (Supplementary Fig. 6a, b) or at longer timepoints (e.g., 10 days or 14 days) (Supplementary Fig. 6c, d). These different choices could alter the landscape of phenopushing hit compounds and targets (Supplementary Fig. 6e–g). Additionally, the platform could have been applied to milder hypoxia conditions (5% $O_2$), where cells show an intermediate degree of hypoxia adaptation after 6 days of exposure (Supplementary Fig. 6h, i).

A fundamental question of any phenotypic screen is whether similarity in phenotypic space translates into similarity of underlying biology and function. In our case, the question is whether pheno-pushed AH cells resemble the protective, adaptive CH state. At the molecular level, we found that downstream components of our top inhibited targets—mTOR and BET proteins—were also inhibited in the CH state. Functionally, we found that inhibiting these target classes conferred functional protection not only for the survival of HepG2 cells (the originally screened cell type) in severe ischemia-like conditions but also for the beating and sarcomere integrity of matured iPSC-CM cells under hypoxic stress. These findings suggest that phenotypically mimicking the adapted state also mimics its protective properties and even translates to other cell types.

By applying our platform to a chemical genetic library, we identified key targets involved in hypoxia adaptation within the druggable target space. Among these, mTOR and BET proteins emerged as top significantly enriched target categories, representing both established mechanisms and less explored aspects of the hypoxia response. mTOR inhibition has been associated with hypoxia-induced translational repression and is primarily studied for its role in mitigating inflammation after hypoxic damage[53–57]. Our findings highlight its ability to accelerate cellular adaptation to hypoxia. Similarly, BET inhibition, known to reduce transcriptional elongation during hypoxia, has been reported only for preventing cell death under hypoxic stress[58,59]. Our results expand its application by demonstrating its role in promoting functional hypoxia adaptation. Expanding the screen beyond the current scope to larger

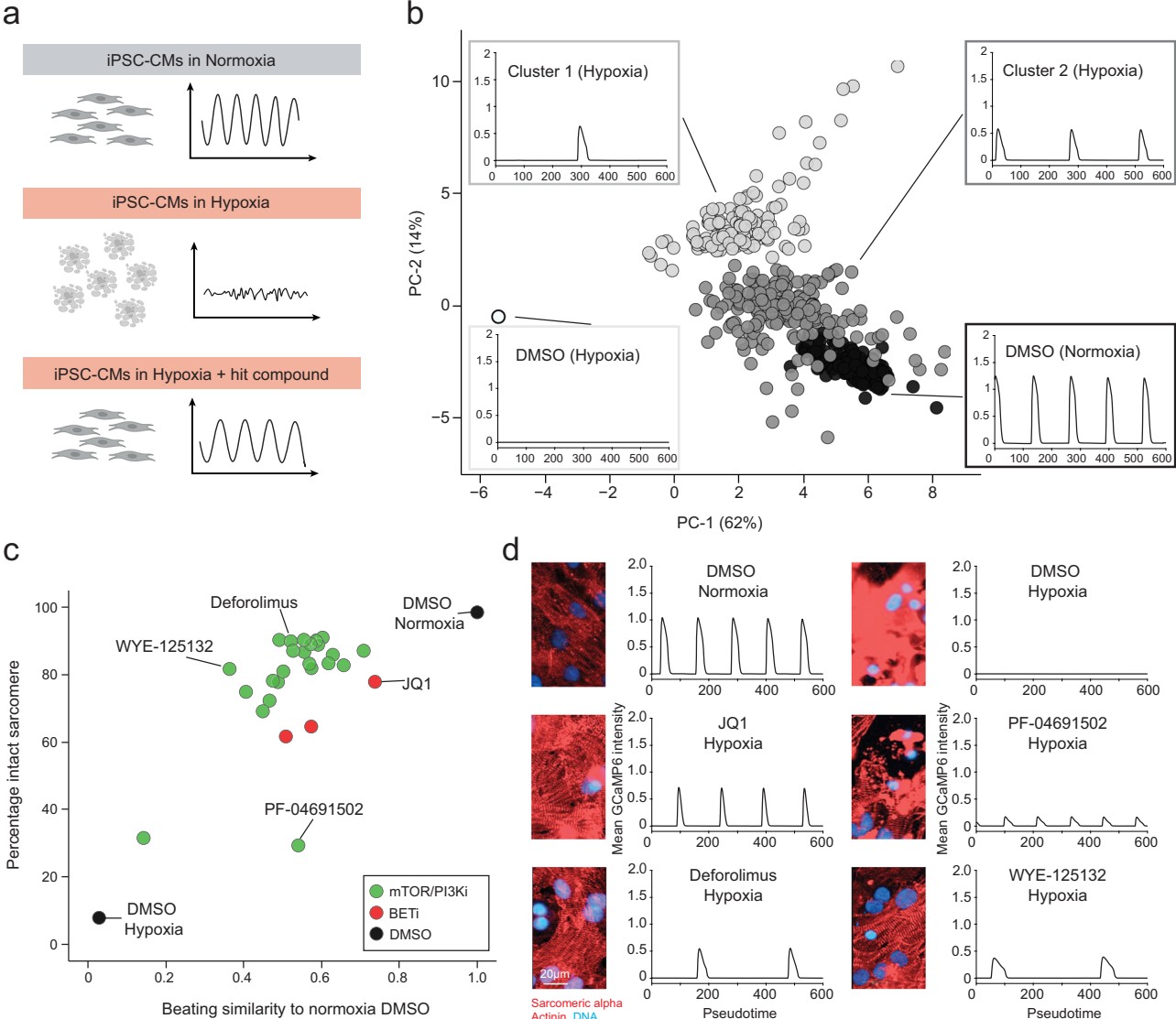

**Fig. 5 | AH-to-CH phenopushing hits rescue iPSC-CMs from acute hypoxic stress. a** Overview of the cardiomyocyte rescue assay. The cell icons and beating pattern icons: Created in BioRender. Wu, A. (2025) https://BioRender.com/q85j552. **b** Survey of phenopushing hit effects on iPSC-CM beating in hypoxia. Beating patterns of iPSC-CMs over the time frame (~20 s) are visualized using principal component analysis. Dot: drug-dose replicate. Colors: (white) 0 beats; (Cluster 1: light grey) 1 beat per time frame; (Cluster 2: dark grey) > 1 beats per time frame; and (black) DMSO normoxia. Representative beating traces for each cluster are shown. **c** Hit compounds in Cluster 2 rescue beating behaviors as well as sarcomere

structures of iPSC-CMs in hypoxia. Scatter plot of beating similarity compared to normoxia DMSO (x-axis) and percentage of intact sarcomere structure (y-axis) for all compounds in Cluster 2 shown in **b**. Dots: average over all replicates for each dose. mTOR/PI3Ki: AH-to-CH phenopushing hits targeting mTOR and/or PI3Ks. BETi: AH-to-CH phenopushing hits targeting BET proteins. **d** Representative immunofluorescence images of sarcomere structure (sarcomeric α-actinin) and corresponding beating traces for selected hits shown in **c**. The experiment was independently repeated three times with similar results. Source data are provided as a Source Data file.

chemical libraries (e.g., diversity-based unknown compound libraries) could uncover additional compounds and targets relevant to hypoxia biology.

Our study opens avenues for future research. To further elucidate the space of possible targets, future screens may make use of expanded chemical libraries, varied oxygen tensions, alternative cell types, or other timepoints at which cells show adaptation (e.g., Supplementary Fig. 6c, d). Additionally, our study focused on compound pre-treatment before the onset of hypoxia; interestingly, many of our hits provided similar phenopushing effects even when treated concurrently with the start of hypoxia (Supplementary Fig. 6j). Future studies may investigate treatment options post-hypoxia insult to identify targets that may be useful in the context of unpredicted hypoxic or ischemic events. Finally, future studies may explore how

well hits found in cellular models translate to animal models and, ultimately, human hypoxia-related diseases. Together, we demonstrate an approach—applicable in situations where biological response networks are complex and incompletely understood—to identify compounds and targets that fast-track cells from stressed to adapted states.

## Methods
### Cell lines
HepG2 cells (obtained from the UCSF Cell Culture Core Facility) were maintained in DMEM with 4.5 g/dL glucose (Thermo Fisher Scientific) supplemented with 10% fetal bovine serum (Gemini Bio #100-106) and 1% penicillin/streptomycin (Thermo Fisher Scientific). Cells were grown in a humidified 37 °C incubator with 5% $CO_2$ for up to 6 weeks,

during which they were passaged every ~4-5 days at ~70–80% confluence with TrypLE (Thermo Fisher Scientific). Cells were exposed to hypoxia by culturing in a glovebox (InvivO₂ 400, Baker Ruskinn) at 1% $O_2$, 5% $CO_2$, and 70% humidity. Aliquots of cell lines were frozen in media with 10% DMSO and stored in liquid nitrogen. All cell lines were routinely tested for mycoplasma as described previously[60,61].

## Compound Library

All compounds were stored at −80 °C as DMSO-dissolved solutions in 384-well PP microplates (Labcyte, #pp-0200). Screened compounds included (1) Selleck FDA-approved & Passed Phase I Drug Library (L3800), (2) Selleck Kinase Inhibitor Library (L1200), (3) Selleck Apoptosis Compound Library (L3300), (4) Selleck Epigenetics Compound Library (L1900), (5) Selleck Bioactive Library (L1700), (6) manually curated FDA-approved & Passed Phase I drugs from other vendors, and (7) manually curated bioactive compounds from other vendors. For a full list of compound details, see Supplementary Data 2.

## Reagents

Phenotypic profiling staining solution was made of 3.24 μM Hoechst 33342 (Invitrogen, H3570), 50 μM BODIPY™ 493/503 (4,4-Difluoro-1,3,5,7,8-Pentamethyl-4-Bora−3a,4a-Diaza-s-Indacene, Invitrogen, D3922), 5.2 μM C11-BODIPY (4,4-Difluoro-1,3,5,7,8-Pentamethyl-4-Bora-3a,4a-Diaza-s-Indacene from Image-iT™ Lipid Peroxidation Kit, Invitrogen, C10445), and 1.9 μM MitoTracker™ Deep Red FM (Invitrogen, M22426) in HBSS. CellTox™ Green Cytotoxicity Assay (G8731) was purchased from Promega. The following antibodies were used according to the dilution factors as primary antibodies in immunofluorescence experiments: HIF-1α (Cell Signaling technology, 36169, 1:200), Phospho-S6 Ribosomal Protein (Ser235/236) (Cell Signaling technology, 4856, 1:1600), Phospho-Akt (Ser473) (Cell Signaling technology, 4060, 1:400), RNA polymerase II CTD repeat YSPTSPS (pSer2) (Abcam, ab193468, 1:100), Phospho-Akt (Thr308) (Cell Signaling technology, 13038, 1:400), Sarcomeric alpha Actinin Monoclonal Antibody (EA-53) (Invitrogen, MA1-22863, 1:200). The following antibodies were used as the secondary antibodies in immunofluorescence experiments: 488-conjugated goat anti-mouse IgG (Invitrogen, A32723, 1:500), 488-conjugated goat anti-rabbit IgG (Invitrogen, A32731, 1:500), 647-conjugated goat anti-mouse IgG (Invitrogen, A32728, 1:500). Alexa Fluor™ 568 Phalloidin (Invitrogen, A12380, 1:400) and Hoechst 33342 (16 μM) were also added to the secondary antibody staining solution for the purpose of cell segmentation.

## Immunofluorescence assay

Cells were seeded in 384-well PhenoPlates (PerkinElmer #6057302) at 2,500 cells per well (an empirically determined density) in 75 μL media and treated with compounds and/or hypoxia as indicated in the respective experiments. Cells were then fixed with 4% paraformaldehyde for 30 min, washed with PBS once, permeabilized with 0.5% Triton X-100 in PBS for 30 mins, and incubated with blocking buffer (0.5% Triton X-100, 2% BSA in PBS) for 1 h. Primary antibodies were used at dilutions indicated in the section antibodies and incubated at 4 °C overnight on a rocker. The next day, cells were washed three times with 0.5% Triton X-100 in PBS and incubated with secondary antibody, Hoechst 33342, and Alexa Fluor™ 568 Phalloidin at room temperature for 1 h on a rocker. Next, cells were washed three times with 0.5% Triton X-100 in PBS. After the final wash, 50 μL of 1× PBS was added to each well. Plates were sealed with adhesive foil and imaged with 20× water immersion lens (NA1.0, effective resolution 0.66 μm) at 5 fields of view, in 3 channels (as indicated in Supplementary Table 1). GraphPad Prism (v9.4.1) was used for graphs and statistical tests (one-way ANOVA and Tukey's correction).

## Imaging and feature extraction

All imaging for the phenotypic profiling of HepG2 cells was performed on the PerkinElmer Operetta CLS System in confocal spinning-disk mode with a 20× water immersion lens (NA1.0, effective resolution 0.66 μm). Each well was imaged with 5 fields of view and 4 channels (filters according to wavelength information in Supplementary Table 1) in 3 z-planes. Cell segmentation and the subsequent single-cell feature extraction were performed through the Harmony™ software (v4.9, Perkin-Elmer) based on maximum intensity projections of each field of view.

## High-dimensional phenotypic profiles for hypoxic response trajectory

In total, 859 features (e.g., intensity, morphology, and texture features, Supplementary Data 1) were extracted from each cell. Phenotypic profiles were constructed on well-level to reflect the feature distributions as previously described[30]. For each well, we compute a Kolmogorov-Smirnov (KS) statistic score for each feature, summarizing the difference between the cumulative distributions of cells in the well and cells from the control condition, defined by pooling all wells in normoxia. Differences in features across batches were assessed using the batch-to-average log fold change for each dimension. Feature categories enriched in the top fold-change dimensions, such as intensity, threshold compactness, radial mean, and axial small, were subsequently filtered out, resulting in 591 features for the subsequent analysis. KS scores for all features were then concatenated to form a phenotypic profile for the selected well. Subsequent analyses (cell state classification, PCA visualization) were based on well-level phenotypic profiles. Computations were performed using Python (v3.9).

## Classification of cellular states based on phenotypic profiles

Cellular state prediction was performed on well-level phenotypic profiles by a k-Nearest Neighbor (kNN) classifier (scikit-learn python package version 1.4.2). 10% of all wells at 5 different hypoxic treatment times (Normoxia, hypoxia 6 h, hypoxia 1 d, hypoxia 2 d, and hypoxia 6 d) were randomly chosen as the test set. The remaining wells were used to train the kNN classifier ($k = 5$) with hypoxic treatment time as the labels. Prediction accuracy of the test wells was calculated by the fraction of correct label assignments for the test set. This procedure was repeated for 100 times and the mean prediction accuracy was used to represent the prediction accuracy of cellular states.

## Measurement of ATP production by Seahorse assay

HepG2 cells were seeded in 96-well Seahorse XFe96 Cell Culture Plates (Agilent, 101085-004) at a density of 10,000 cells per well and treated with compounds ~1 h later. Metabolic activity was measured using an Agilent Seahorse XFe96 Analyzer following the manufacturer's protocol for Mito Stress Test and analyzed using the manufacturer-provided online analysis tool (https://seahorseanalytics.agilent.com/).

## AH-to-CH phenopushing compound library screen

The primary compound screen was performed in multiple batches (Supplementary Table 2). Each batch contained eight screening plates plus three plates for batch-level QC: a normoxia control plate (DMSO only), a 6 d CH control plate (DMSO only) for QC assessment of hypoxia responses, and a 24 h AH reference compound plate (Supplementary Fig. 4a) for QC assessment of cellular responses to compounds in AH. For plate-level QC, each screening plate also included positive control wells (with bioactive compounds) and negative control wells (DMSO).

Within each batch, our screen workflow was designed to seed cells in all plates at the same time as well as to stain/fix cells at the same time. HepG2 cells were split from flasks and seeded into 384-well PhenoPlates (Perkin-Elmer) at ~2500 cells per well (~3500 cells for the 6 d hypoxia condition to account for growth rate differences) in 75 μL of media and treated with compound for 48 h before cell staining.

For chronic hypoxia plates, we started with cells from a flask that had already been treated with 1% $O_2$ for 4 d. Cells were then plated into the 384-well plate using media that had been preconditioned

with 1% $O_2$. All other plates were seeded with cells maintained in normoxia. Right after plate seeding, compounds were added with a final concentration of 0.1% DMSO. The compound addition was performed with the ECHO 650 liquid handling system (Beckman Coulter, Brea, CA) controlled through a Perkin Elmer EXPLORER G3 WORKSTATION. Screening compounds were tested in duplicate at 2 doses (high dose of 10 μM or 2 μM and a 10-fold dilution). Because the EXPLORER G3 workstation was also integrated with a 1% $O_2$ incubator, exposure to ambient air was minimized for chronic hypoxia plates except for the brief duration of ECHO-mediated compound addition to the plate.

After compound addition: (1) the 6 d CH control plate was immediately transferred to the hypoxia glovebox, (2) the normoxia control plate was returned to the normoxia incubator, and (3) the 24 h AH reference compound plate and eight screening plates were incubated in normoxia for 24 h before transfer to the hypoxia (1% $O_2$) glovebox for another 24 h. After a total of 48 h compound treatment, cells were stained by adding 10 μL of a freshly prepared 8× dye master mix in pre-warmed media (see staining reagents) and incubated at 37 °C for 1 h in their corresponding incubators (N plates in normoxic incubator, AH and CH plates in hypoxic incubator). Cells were then fixed by adding 30 μL 16% paraformaldehyde for 30 min at room temperature. Finally, cells were washed three times with 1× HBSS, sealed with adhesive foil for light-protection until imaging. All pipetting steps were performed by MultiFloFX and 405 TS washer (BioTek, Agilent Technologies) controlled through the Perkin Elmer EXPLORER G3 WORKSTATION automation system. Plate handling steps were established using PerkinElmer's plate::works™ (v6.2) software.

The experimental workflow in the confirmation screen was identical to the primary screen, except compounds were tested at 6 concentrations (10 μM, 2 μM, 0.4 μM, 0.08 μM, 0.016 μM, and 0.003 μM) in 3 well replicates.

### High-dimensional phenotypic profiles for compound screening

Phenotypic profiles were constructed on well-level similar as described in earlier section (High-dimensional phenotypic profiles for hypoxic response trajectory). For the purpose of identifying AH-to-CH phenopushing hits, the phenotypic profiles were calculated using AH DMSO condition as the control. Phenotypic profiles were calculated batch-wise: For a specific feature, the difference in cumulative distribution functions (CDF) between cells in a selected well and cells from control condition (pooled cells from all DMSO-treated wells in AH within this batch) were summarized by a Kolmogorov-Smirnov (KS) statistic. KS scores for all features were then concatenated to form a phenotypic profile for the selected well. Subsequent analyses (QC, AH-to-CH phenopushing hit calling) were based on well-level phenotypic profiles.

After summarization of well-level phenotypic profiles, feature selection was performed for batch bias removal. We first calculated PCA embeddings based on combining all well-level phenotypic profiles from all batches. Among the top 5 PCs, we identified the PC dimension that accounted for the most variance. We further checked the feature loading scores for PC2 and identified the top 4 feature types with the strongest batch effects and removed them for subsequent analysis. Computations were performed using Python (v3.9).

### Quality control for compound screening

Quality control for compound screening was performed on both batch-level and plate-level.

**Batch-level QC.** Separability between normoxia, acute hypoxic and chronic hypoxic states (represented by DMSO-treated wells in each condition) as well as the classification accuracy of compounds within a reference library was used as quality control for each batch. (1) Cellular state predictions using phenotypic profiles of DMSO-treated wells was performed on well-level phenotypic profiles by k-Nearest Neighbor

(kNN) classifier in the same procedure as described in the Classification of cellular states based on phenotypic profiles section (Supplementary Fig. 4b). (2) Compound category classification for reference compound set (Supplementary Data 2) was performed on well-level phenotypic profiles in the same procedure as cell state classification, except that the phenotypic profiles of selected compound wells were first transformed by linear discriminant analysis with category names as the labels before kNN classification. (Supplementary Fig. 4c)

**Plate-level QC.** The plate-level QC was performed by analyzing the phenotypic profiles of control wells (DMSO wells as negative control, AZD8055 or Fluvastatin treated wells as bioactive control) on AH plates. (1) For each batch, compound classification was performed on DMSO and bioactive (AZD8055, Fluvastatin) control wells in all AH plates. (Supplementary Fig. 4d). (2) The Mahalanobis distance of the phenotypic profiles of each DMSO control wells to the centroid of the pooled AH DMSO control wells within the same batch were calculated (as indicated later in the section AH-to-CH phenopushing hit-calling for primary screen) and compared for plate-level variability. (Supplementary Fig. 4e)

### AH-to-CH phenopushing hit-calling for primary screen

Hit calling was performed as in the following steps:

Identification of wells that move out of the AH DMSO cloud. We calculated if there was detectable alteration in phenotypic profiles of screening wells from the control wells (DMSO-treated wells in AH condition within the same batch) batch-wise. The Mahalanobis distance of the phenotypic profile of each screening well relative to the centroid of control wells were calculated. The Mahalanobis distance of each control well towards control centroid was also calculated to build up the control background distribution. The Mahalanobis distance of a given well was compared with this background distribution. Screening wells with a significantly larger Mahalanobis distance (t-test, p-value < $10^{-6}$) were identified as bioactive wells.

Distance-based AH-to-CH phenopushing calculation and hit calling. DMSO-treated wells in CH condition across all batches were combined as the destination cell state (CH cloud) for AH-to-CH phenopushing. For well i, the Mahalanobis distance ($D_i$) of each screening well was calculated as the distance from its phenotypic profile to the centroid of the destination cell state ($C_{CH}$). As reference, the $D_i$ of DMSO-treated wells in N, AH and CH conditions were also calculated. The $D_i$ of each screening well i was normalized to acute hypoxia DMSO-treated wells within the same screening plate by Z-score: $ZScore_i = (D_i - \bar{D}_{DMSO}) / \sigma_{DMSO}$, where $\bar{D}_{DMSO}$ and $\sigma_{DMSO}$ represent the mean value and standard deviation of the $D_i$ of AH DMSO wells on the same screening plate. A tested perturbation is identified as hit perturbation if at least one of the duplicate wells is bioactive (bioactivity p-value < $10^{-6}$) and has $ZScore_i \leq -3$ (where lower $ZScore_i$ is "desirable", i.e., indicates a closer distance towards the chronic hypoxia centroid).

Direction-based AH-to-CH phenopushing calculation and hit calling. The AH-CH reference vector was defined as the vector from the centroid of pooled DMSO wells in AH from all batches ($C_{AH}$) to the centroid of pooled DMSO wells in CH from all batches ($C_{CH}$): $V_{CH\_AH} = C_{CH} - C_{AH}$. For each well i (both screening wells and DMSO-treated wells in N, AH, CH conditions), the cosine similarity between its pheno-shift vector from $C_{AH}$ ($V_i$) and the AH-CH reference vector ($V_{CH\_AH}$) was calculated and converted to an angle ($\theta_i$): $\theta_i = \cos^{-1}(V_{CH\_AH} \cdot V_i / \| V_{CH\_AH} \| \|V_i\|)$. The $\theta_i$ of all DMSO-treated wells in CH condition ($\theta_{CH}$) across all batches were combined and ranked from smallest to largest to form a positive control distribution of angles. A tested perturbation is identified as hit perturbation if at least one of the duplicate wells is bioactive (bioactivity p-value < $10^{-6}$) and has $\theta_i$ within the top 90% tile of the $\theta_{CH}$ population ($\theta_i \leq P_{90}(\theta_{CH})$).

Hit perturbations were then matched onto compounds. The union of hit compounds identified from distance- and/or direction-based hit-calling constituted the primary phenopushing hit compounds.

### Hit-calling for multi-dose confirmation screen

Bioactivity was calculated using the DMSO wells (negative control) under acute hypoxia within the corresponding batch. For phenopushing calculations, the destination cell state was defined by combining DMSO wells under chronic hypoxia from both the primary screen and the confirmation screen. The $D_i$ and $\theta_i$ for all test wells and DMSO-treated wells at N, AH, CH conditions were calculated. The same AH-to-CH phenopushing hit-calling cutoff was used as described in the Hit-calling for primary screen section. A perturbation (compound + dose) was identified as a hit perturbation if at least 2 out of 3 well replicates met the hit criteria. The list of confirmed phenopushing hit compounds reflected all compounds with hit perturbations at any dose.

### Target overrepresentation analysis

Target profiles for screening compounds were summarized based on compound activities in the ChEMBL database (version: ChEMBL32) and then subjected to the following criteria: specific target type (single protein, protein complex), organism (Human), standard type ($IC_{50}$, $EC_{50}$, $K_d$), and more than 3 reported activity observations. For each compound-target pair, the median value of all reported observations was calculated as the activity value. For each screened compound, the targets with median $P_{activity}$ (representing the $-\log_{10}$ of the median activity values) $\geq 6$ were curated as the active targets. Each target was then mapped onto the screened compounds for that target. We applied hypergeometric distribution overrepresentation analysis for each target's association with phenopushing hits. The corresponding P-values were adjusted for false discovery rate (Benjamini–Hochberg procedure). Targets with an adjusted P-value $\leq 0.05$ were considered overrepresented targets for phenopushing.

### Hit rate vs potency calculation

Tested compounds for a specific target were separated into different potency ranges based on their activity values (see "Target overrepresentation analysis"). For BET proteins, the median activity over multiple BRD isoforms (*BRD2, BRD3, BRD4, BRDT*) was used. For PI3Ks, the median activity over all reported PI3K isoforms (*PIK3C2A, PIK3C2B, PIK3C3, PIK3CA, PIK3CB, PIK3CD, PIK3CG, PIK3CG/ PIK3R5, PIK3R1/PIK3CA, PIK3R1/PIK3CB, PIK3R1/PIK3CD, PIK3R1/PIK3CG*) was used. Hit rate for a specific target within a potency range was calculated as the percentage of confirmed hits identified (from either distance-based or direction-based hit-calling) among the subset of screened compounds whose activity value fell within the potency range for that target (Fig. 3b–d).

### Hit evaluation in ischemia-like conditions

HepG2 cells were seeded into 384-well PhenoPlates (Perkin Elmer) at an empirically determined density (2500 cells/well) in 75 μL of media per well. Compound additions were performed using the same protocol as in the 'AH-to-CH phenopushing compound library screen' section. Compounds were tested at 3 doses (10 μM, 1 μM, 0.33 μM) in 4 replicates. After 48 hr of compound treatment, media was replaced with media containing CellTox™ Green according to the manufacturers protocol. Cells were then subjected to hypoxia (1% $O_2$) with deprivation of glucose and FBS (to mimic ischemia-like stress) or without nutrient deprivation. The duration of exposure to ischemia-like stress was calibrated so that only ~10% of the DMSO cells remained (~48-72 h in ischemic conditions, Fig. 4b). Next, cells were washed and fixed with paraformaldehyde (at a final concentration of 4%) for 30 min at room temperature. Plates were then stained with Hoechst 33342, washed with 1x HBSS, and sealed with adhesive foil for light-protection until imaging. Imaging was performed on the Operetta CLS confocal spinning-disk high-content analysis system (Perkin Elmer) using a 10x objective and appropriate filter settings (Supplementary Table 1). The full area of each well was imaged. Cell segmentation and subsequent single-cell feature extraction were performed using the Harmony™ (v4.9) software. Live-cell count was calculated by applying a threshold

to the mean nuclear CellTox™ Green intensity. Cell survival was normalized to the mean survival of DMSO-treated cells.

### Human iPSC-derived cardiomyocyte differentiation

The GCaMP6 and inducible dCas9-KRAB expressing iPSC line in the WTC background was obtained from the Gladstone Institute Stem Cell Core Facility. Human induced pluripotent stem cell (iPSC)-derived cardiomyocytes (iPSC-CMs) were generated as described previously by combining published protocols[47–49,62]. Briefly, TrypLE (Thermo Fisher Scientific) was used to dissociate iPSCs into a single-cell suspension, then 75k of iPSCs were seeded to each well on a 12-well plate that was coated with Geltrex™ LDEV-Free Reduced Growth Factor Basement Membrane Matrix (Thermo Fisher Scientific). Cells were plated in E8 medium (Thermo Fisher Scientific) supplemented with 10 μM Y-27632, and daily medium changes to E8 (without Y-27632) was performed thereafter. Differentiation was initiated when the cells were at 80–90% confluence (approximately 48 hours post-seeding) by switching to RPMI-1640 medium supplemented with B27 (minus insulin; Thermo Fisher Scientific) and 7.5 μM CHIR99021. 48 h later, the cell culture medium was sequentially changed every 48 hours: first to RPMI-1640 with B27 (without insulin) and 7.5 μM IWP2; then to RPMI-1640 with B27 (without insulin); and finally, to RPMI-1640 with B27 containing insulin and 1% penicillin/streptomycin, which was refreshed every 2-3 days. The spontaneous beating could be observed 8–10 days after initiation of differentiation. Spontaneously beating iPSC-CMs (>80% of cells beating) were enriched for 6 days in lactate enrichment medium and reseeded into 75 cm² flasks coated with Geltrex (at a seeding ratio of one 12-well plate to one flask) in RPMI-1640 with B27 (plus insulin) and 2 μM CHIR99021. Once the spontaneously beating iPSC-CMs reached confluency, they were replated into 384-well PhenoPlates (at a seeding ratio of one flask to one 384-well plate) coated with Geltrex and maintained in maturation medium for 10 days. These matured iPSC-CMs were used for all subsequent experiments.

### Hit evaluation using iPSC-derived cardiomyocytes

Compounds were added directly to each well of matured iPSC-CMs in 384-well Phenoplate's (Perkin Elmer) and incubated at 1% oxygen in a glovebox (InvivO₂ 400, Baker Ruskinn) 48 h after drug addition. Compounds were tested in 4 replicates. Once cardiomyocyte beating was lost in most DMSO wells (~48-72 h in hypoxia), GFP signal (reflecting $Ca^{2+}$ transients) was monitored using a EVOS M7000 Imaging System set at 37°C, 80% humidity, 5% $CO_2$ and 1% $O_2$ (Thermo Fisher Scientific) for 30 seconds at a frame rate of 30 Hz in the GFP channel (excitation max 488 nm) using a 20x objective (one FoV) for a total processing time of ~12 h. Normoxic iPSC-CMs from the same batch of cells was imaged after the hypoxic experiments were complete. Subsequently, cells were fixed with paraformaldehyde (final concentration 4%) for 30 min at room temperature. To assess intactness of sarcomere structure, immunofluorescence staining was performed as described above using a sarcomeric α-actinin monoclonal antibody (EA-53, Invitrogen #MA1-22863). In addition, the actin cytoskeleton was stained using phalloidin. Plates were then sealed with adhesive foil and light-protected until immunofluorescence confocal imaging on the Operetta CLS (Perkin Elmer) at 20×. Each well was imaged at 5 fields of view, in 3 channels (filters according to wavelength information in Supplementary Table 1) in 3 planes.

### Analysis of iPSC-derived cardiomyocyte $Ca^{2+}$ transients

Mean fluorescence intensities for each field of view were extracted by the scikit-image (v1.2.2) module. The signal was baseline-corrected by the Python package BaselineRemoval (v0.1.3). For each compound, the mean signal intensity of frames 250-850 (~20 sec) was plotted. Peaks were detected from the resulting time series data and converted into peak feature vectors using a custom analysis pipeline in Python. All extracted beating features are listed in Supplementary Table 3. The

similarity of beating patterns was quantified by calculating the Manhattan distance of replicate-averaged peak feature vectors compared to DMSO-treated iPSC-CMs in normoxia and min-max scaled.

## Analysis of iPSC-derived cardiomyocyte sarcomere structure

Subsets of image pixels were manually labeled as either background (no cells), intact sarcomere structure, or deteriorated sarcomere structure using the Python package Label Studio (v1.9). Image features of the labeled regions were extracted using the scikit-image (v1.2.2) module "multiscale_basic_features" and used to train a random forest classifier. New images were then subjected to the same feature extraction procedure, and the trained random forest classifier was used to predict the labels for all pixels. The resulting predicted labels were manually spot checked. The percentage of all imaged pixels labeled as intact sarcomere was used for all downstream analysis.

## Comparison of transcriptomics data from GTEx healthy human tissues and CCLE cancer cell lines

The Genotype-Tissue Expression (GTEx) and Cancer Cell Line Encyclopedia (CCLE) RNA-seq datasets, comprising 53 healthy human tissues and 934 cancer cell lines, respectively, were downloaded as FPKM values from EMBL-EBI database[22,23]. First, the expression data were scaled to ensure consistent library sizes. Next, gene expression was Z-score transformed across all cell lines (for CCLE data) or tissues (for GTEx data). Then, the top 200 expressed genes were identified in each cell line or tissue, and the number of overlapping genes between cell lines or tissues was used to measure their similarity. Pairwise comparisons were made between each tissue and each cell line (i.e., 49,502 comparisons) to determine the number of shared top 200 expressed genes (Supplementary Fig. 1). The best-matching tissue-cell line pair (defined as the highest number of shared top 200 expressed genes) was liver and HepG2. The same pair was identified when examining top 500 or top 1000 expressed genes.

## RNA-seq and data processing

HepG2 cells were cultured in 6-well plates and treated with hypoxia for the indicated time. Approximately one million cells were collected by trypsinizing. The cell pellet was washed with PBS and stored at −80 °C until extraction with RNeasy Plus Mini kit (Qiagen #74134) according to manufacturer instructions. Preparation of sequencing libraries (NEBNext Ultra II RNA Library Prep Kit), sequencing (Illumina HiSeq 2 × 150 bp paired end), and demultiplexing (Illumina bcl2fastq v2.20) were conducted by Azenta Life Sciences (Genewiz). After mapping reads to GRCh38 using STAR (version 2.7.9a), the count matrix was obtained using featureCounts (version 2.0.1) with reference gene annotations from Gencode v38. Count data were normalized using DESeq2 (version 1.44.0). For differential expression analysis, DESeq2 was used to compare hypoxia conditions against normoxia. Log2 fold change (log2FC) and adjusted p-values (padj) were obtained for each gene in each comparison. PCA analysis was performed with median-normalized read counts scaled using StandardScalar in scikit-learn (version 1.5.2) (Supplementary Fig. 2a). Pathway enrichment was performed using Gprofiler (version 1.0) on genes that have $|\log_2FC| > 1.5$ and padj<0.05 (Supplementary Fig. 2b). The RNAseq data were deposited in the Gene Expression Omnibus (GEO) repository (accession number: GSE283995).

## Statistics and reproducibility

**Replicates in screening.** No statistical method was used to predetermine sample size. At least three biological replicates were included in all experiments, except in the primary AH-to-CH phenopushing screening, in which two biological replicates were included. All findings were successfully repeated in independent experiments.

**Replicates in non-screening experiments.** No statistical method was used to predetermine sample size. 48-128 biological (well) replicates

were used to define the hypoxia response trajectory by phenotypic profiling (Fig. 1b, Supplementary Fig. 6a, c, i). 6 biological replicates were included in detection of HIF-1α accumulation (Fig. 1d). 6 biological replicates were included in detection of metabolism shift (Fig. 1e). At least 10 biological replicates were included in the detection of the protective effect of chronic hypoxia in ischemic conditions (Fig. 1f, Supplementary Fig. 6d). For the annotated compound library used in the phenopushing screen (Fig. 2), the number of compounds curated (6011) was based on target coverage. 2 well replicates X 2 doses were included for each tested compound in the primary screen. 3 well replicates X 6 doses were included for primary hits in the validation screen. At least 4 biological replicates were included in the study of compound hits' downstream mechanisms (Fig. 3e–g, Supplementary Fig. 5a, Supplementary Fig. 6h) and functional protections (Figs. 4 and 5). All experiments were repeated at least 3 times with similar results.

**Data exclusion.** No data were excluded from the analyses.

**Randomization.** When multiple compounds were tested within one experiment (Figs. 2–5), compounds were randomized on the plates instead of being grouped by mechanism of action. Within each experiment, plates were stained and imaged in a randomized order.

**Blinding.** All data collections were performed without knowing the labels of the samples. Within each experiment, data analysis was first performed with only the group labels (negative control or test). The exact metadata, such as the compound name and target information, were added from plate maps after all analysis was completed.

## Reporting summary

Further information on research design is available in the Nature Portfolio Reporting Summary linked to this article.

## Data availability

Processed phenotypic profiling data relevant to each figure has been deposited in CSV format on Zenodo at the following link: https://doi.org/10.5281/zenodo.14837176. RNA-seq data from GTEx and CCLE FPKM datasets can be accessed from the EMBL-EBI database under the accession numbers E-MTAB-5214 and E-MTAB-2770, respectively. RNA-seq data generated in this study is deposited in GEO repository (GSE283995) at the following link: https://www.ncbi.nlm.nih.gov/geo/query/acc.cgi?acc=GSE283995. Source data for generating figures are provided as a Source Data file with this paper. Source data are provided with this paper.

## Code availability

The code necessary to replicate the results has been made available in Jupyter Notebooks and can be accessed on Zenodo at the following link: https://doi.org/10.5281/zenodo.14837176.

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

## Acknowledgements

We are grateful to Tristan McClure-Begley, Matthew Jacobson, and Jack Taunton, as well as Christopher Waters, Carolyn Ku, and other members of the Altschuler and Wu laboratories for their suggestions and feedback. We acknowledge the following financial support: DARPA Panacea program HR0011-19-2-0018 (L.F.W. and S.J.A.), Human Frontiers Science Program LT000908/2020-C (L.L.), National Institute on Aging R38AG070171 (S.Q.S.), and National Institute of Mental Health R25MH0602 (S.Q.S.).

## Author contributions

S.J.A. and L.F.W. designed, conceptualized, and supported the study. L.L., H.H., S.Q.S., and S.H. performed the experiment and collected the data. L.L., H.H., S.Q.S. and F.B. analyzed and interpreted the data. L.L., H.H., S.Q.S., S.J.A. and L.F.W. wrote the paper. All authors have given approval for the final version of the paper for submission.

## Competing interests

S.J.A., L.F.W., L.L., H.H., S.Q.S., S.H., F.B. are coinventors of US Application No. 63/559,109, a patent application related to the subject matter of this publication, assigned to The Regents of the University of California.
