## [Transparent Peer Review file · Nature Communications]

A phenopushing platform to identify compounds that alleviate acute hypoxic stress by fast-tracking cellular adaptation

Corresponding Author: Dr Steven Altschuler

Version 0:

Reviewer comments:

Reviewer #1

(Remarks to the Author)

Li et al explore adaptation to hypoxic stress using phenotypic responses and a compound screen. They develop a platform to measure typical phenotypical cell responses to acute or chronic hypoxia (defined from 6 hr to 6 days). Having established these phenotypes they use a compound library screen to find compounds that promote phenotypes consistent with an adaptive response to hypoxia. They go on to validate the top hits and demonstrate their utility in iPSC-cardiomyocytes. Overall, this is a very interesting study with high quality experiments. The biological question is important, and the authors rightly highlight that other oxygen sensing pathways aside from HIF must be important for adaptation to hypoxia. I also found the approach novel and applicable to examine other stress responses, aside from hypoxia. I have a few minor comments/suggestions below, but otherwise I think the work presented is of very high quality.

- 1) The phenotypic changes between AH and CH are clear, but 6 hrs is still a relatively long period of hypoxia, and certainly in considering ischaemia. Are phenotypic changes observed over shorter time periods?
- 2) The authors focus on lipid, MitoTracker and Hoechst staining but not transcriptional adaptation. Given that most of our current knowledge regarding adaptation to hypoxia relates to HIFs (and to a lesser extent epigenetic marks) are HIFs important for the phenotypes they observe? HIF depletion may help address this. Additionally, are there any associated transcriptional phenotypes alongside the morphological/lipid changes that they characterise?
- 3) The authors chose 1% oxygen for all their studies but if there is an adaptive response it would be expected to be graded along with oxygen availability. Are similar responses observed at other oxygen concentrations? For instance, is there a similar adaptive effect with mTOR/PI3Ki at <1% oxygen or 5% oxygen in the iPSCs or HepG2s?

(Remarks on code availability)

I am not a computational biologist and am not able to provide an expert opinion on the suitability of the models generated for the high dimensional phenotypic profiles. They are clearly described in the methods and code made available.

Reviewer #2

(Remarks to the Author)

The article by Li et al. presents a novel platform for high-content phenotypic screening to identify bioactive compounds that can accelerate cell adaptation to hypoxia, with the goal of alleviating acute hypoxic insult. Knowing that acute hypoxic stress can be detrimental to cells, leading to their irreversible damage and ultimately cell death, and that hypoxia preconditioning is one of the methods for cell adaptation to acute hypoxia-related stress, the authors hypothesized that shifting cells from a severe acute hypoxic state to a milder chronic state may provide a strategy to rescue cells from damage or death. The authors developed a microscopy-based, high-dimensional screening method designed to distinguish different cell states exposed to different oxygen conditions, ranging from normoxia (atmospheric oxygen levels) to acute hypoxia (1% oxygen for 6-48 hours) to chronic hypoxia (1% oxygen for 6 days). Using this platform and a large compound library, they found that inhibition of mTOR or BET proteins in particular provides significant cytoprotection against ischemic stress.

Overall, the study is solid and interesting from both a methodological and biological point of view. In terms of methods, the authors relied on high-throughput fluorescence microscopy and bioinformatics to follow the changes in different cell phenotypes. Thus, this platform appears to be useful for applications aimed at investigating other stress-related conditions in cells. From a biological perspective, the study provides new insight into hypoxia-mediated changes in cells that could be modulated by inhibiting specific intracellular pathways. However, the authors used a cancer cell line to screen for stress-induced phenotypes. Since cancer cells differ from normal cells in their response to hypoxia, e.g. in terms of cell cycle or DNA damage, the presented results may not fully reflect the cellular milieu of normal cells. Thus, the compound screening data may be more appropriate for studies in cancer-related research, but not for any cell type. This should be better highlighted in the limitations of the study, as some phenotypes occurring in normal cells may not have been captured. In my opinion, the limitations are described too briefly. However, it must be acknowledged that the authors also used an iPSC-derived model of oxidative damage in cardiomyocytes that showed beneficial effects of mTOR or BET inhibition, which is consistent with the data from HepG2 cells.

The analytical approach is sound and comprehensive. The methods are described in sufficient detail. Results are presented clearly and data analysis is performed accurately. There are only a few suggestions to consider:

1. The authors could elaborate a bit more on the advantages/disadvantages of the image-based phenotypic profiling presented in this study over other approaches to cell fate analysis, such as proteomic/kinetomic/metabolomic. Could the authors add more on this point in the Introduction or Discussion?
2. The introduction is rather scarce. It could include more details on canonical and non-canonical mechanisms of hypoxia sensing in cells. This would help readers to better understand the background of the study.
3. If I am not mistaken, the code for comparing transcriptomic data for cancer cell lines and tissues is not provided.
4. What post hoc test was used to calculate statistical significance after ANOVA? It is not specified in the text or figure legends (Figure 1, Figure 3, Extended Data Figure 3).
5. Why did the authors select 6 h of hypoxia treatment as the starting point for their analyses? It is known that hypoxia induces immediate changes in the activity of some proteins and in cellular metabolism. Please comment on this in your manuscript.
6. Extended data Figure 2 - the font size is too small. Please increase it, especially for the names of the color-coded pathways shown on the right.
7. "Fig. 2e" in the text (line 120) should be Fig. 2d.
8. The authors state that: "Compounds associated with mTOR/PI3K and BETs were most effective in rescuing iPSC-CM's from hypoxic stress" [...] (lines 201-202), suggesting that other compounds were also tested. However, Figure 5C shows that only mTOR/PI3K and BETs inhibitors were used (although the legend in the scatter plot also indicates the use of other inhibitors in gray).

(Remarks on code availability)

The code and data descriptions are clear and easily accessible. However, I did not review the code in detail. The authors have provided a README file with sufficient information about installation requirements.

Reviewer #3

(Remarks to the Author)

The manuscript from Li et al describes the development of a high-content platform to screen for molecules that are able to facilitate adaptation to hypoxic stress. This approach was utilized to identify such compounds, which were found to largely involve mTOR/PI3K or BET inhibition activities, and were able to promote the survival of both liver cells exposed to ischemia-like stress and rescue cardiomyocytes from hypoxia stress. This work demonstrates a "ground-up" approach for assessing effectors of cellular phenotypes independent of target, and provides new means for identifying molecules with desired activities and new targets associated with biological processes, in this case hypoxia response and drivers to chronically adapted cellular states. The authors were thorough and did a nice job of presenting evidence to support their hypotheses. It is also appreciated that the compounds' effects were evaluated in two different cell types. The studies performed and results obtained are quite significant and merit publication in Nature Communications. There are some points that would be useful for the authors to address to improve the manuscript. Please find additional information in this regard in the comments below.

General/major comments:

- 1.) The authors should provide additional information regarding hypoxia adaptation responses in the introduction. It is only mentioned that there has been a focus on HIF/PHD/VHL in the introduction. At the very end (in the discussion), it is indicated that mTOR inhibitors and BET inhibition have been used to alleviate hypoxia-related stress. As written, prior knowledge of the mTOR/PI3K/BET pathways' involvement and novelty of the findings presented are unclear.
- 2.) The authors elected to use 6-day exposure to hypoxia as their model for chronic (adapted) hypoxia (and hence benchmark for compound evaluation). Was any longer period of time examined? It would be useful to know whether the 6-day mark was in fact adapted to the greatest extent possible or if it is only partially adapted. If in fact, longer than 6-days resulted in a "more adapted" state, would the number of hits (and their target distribution) change (i.e., which compounds would meet the new benchmark)?
- 3.) The authors should be clearer about the timing of compound treatment within the results section; the diagram in Figure 2a

is a bit confusing. As written in the methods, it seems that the compounds are being used prior to initiation of hypoxia. It would be important to determine whether the molecules can initiate phenopushing in cells following the onset of hypoxia as well.

4.) Having found that mTOR/PI3K/BET members were molecular targets of "hit" compounds, the authors assessed the down-regulation of these proteins' activity across hypoxic states (Figures 3e-g). Can the authors provide some rationale for why deactivation might result in CH? This is especially interesting since pS6 and pAkt levels in CH are closer to those of AH, and in all three cases, normoxia has the highest levels, followed by AH and then CH.

Specific/minor comments:

1.) It is very difficult to distinguish the circles used for 1 and 10 uM in Figure 4c. Can a mixture of colors and sizes (or some other approach) be used to make the figure clearer?

2.) Lines 220-221 – "...to identify for compounds that push cell..." should be revised to read "...to identify compounds that push cells..."

3.) In line 236, the authors discuss limitations of the study, which this reviewer does not believe is necessary. However, it would be useful if the authors utilized the discussion to present directions for further studies to study CH and these compounds.

4.) Line 323 – "...time..." should be "...times..."

(Remarks on code availability)

Version 1:

Reviewer comments:

Reviewer #1

(Remarks to the Author)

The authors have fully addressed all the concerns raised.

(Remarks on code availability)

Reviewer #2

(Remarks to the Author)

I would like to thank the Authors for addressing all my concerns and recommendations. I have no further requests.

(Remarks on code availability)

Reviewer #3

(Remarks to the Author)

This reviewer thanks the authors for their thoughtful responses and additional results presented in the present version of the manuscript. The only remark I have is the following:

The authors present new data in the current revision (Extended Data Fig 6 a-i), however their descriptions are limited to brief statements within the Discussion Section. It is recommended that these results be presented accordingly in the Results Section, with synopses/conclusions from them in the Discussion section.

(Remarks on code availability)

Summary:

We are grateful to the reviewers for their constructive feedback. We have revised the manuscript and added data to address points raised by the reviewers. The suggestions from all reviewers have greatly helped to improve the clarity and claims of our manuscript.

Below is a point-to-point response.

Reviewer 1:

Li et al explore adaptation to hypoxic stress using phenotypic responses and a compound screen. They develop a platform to measure typical phenotypic cell responses to acute or chronic hypoxia (defined from 6 hr to 6 days). Having established these phenotypes they use a compound library screen to find compounds that promote phenotypes consistent with an adaptive response to hypoxia. They go on to validate the top hits and demonstrate their utility in iPSC-cardiomyocytes.

Overall, this is a very interesting study with high quality experiments. The biological question is important, and the authors rightly highlight that other oxygen sensing pathways aside from HIF must be important for adaptation to hypoxia. I also found the approach novel and applicable to examine other stress responses, aside from hypoxia. I have a few minor comments/suggestions below, but otherwise I think the work presented is of very high quality.

1) The phenotypic changes between AH and CH are clear, but 6 hrs is still a relatively long period of hypoxia, and certainly in considering ischaemia. Are phenotypic changes observed over shorter time periods?

As requested, we profiled earlier time points. We do observe phenotypic changes at earlier timepoints of 2h and 4h, though the differences among these early timepoints (2, 4, and 6hrs) are somewhat subtle. We have added these new data to the manuscript (Discussion lines 230-232, **Extended Data Figs. 6a, b**).

2) The authors focus on lipid, MitoTracker and Hoechst staining but not transcriptional adaptation. Given that most of our current knowledge regarding adaptation to hypoxia relates to HIFs (and to a lesser extent epigenetic marks) are HIFs important for the phenotypes they observe? HIF depletion may help address this. Additionally, are there any associated transcriptional phenotypes alongside the morphological/lipid changes that they characterise?

To assess transcriptional response to hypoxia, we performed RNA sequencing of HepG2. Transcriptional profiles showed distinct temporal changes mirroring the image-based findings (**Extended Data Fig. 2a**), with HIF1 signaling peaking in the acute hypoxic state and remaining enriched but at lower levels in the chronic hypoxic state (**Extended Data Fig. 2b**). We also observed that pathways related to lipid metabolism and cell morphology are enriched, consistent with phenotypic changes observed in our image-based screen (**Extended Data Fig. 2b**). These data are now added as **Extended Data Fig. 2** and to lines 91-92 of the manuscript.

To assess the role of HIFs in the observed phenotypic response, we focused on HIF1 α , a key player in short-term hypoxia response. We generated a HIF1 α knockout HepG2 cell line. Interestingly, WT and KO cells had somewhat similar response trajectories. This suggests that HIF1 α is not the sole driver of the observed phenotypes in our assay. A more comprehensive investigation, outside the scope of this study, is needed to pinpoint the combination of HIFs and/or

other oxygen-dependent pathways contributing to the response. The HIF1 α KO data is provided below for the reviewer.

Response Figure 1. Comparison of HIF1 α KO vs WT HepG2 cells in hypoxia response. (a) Induction of nuclear HIF1 α accumulation in HepG2-WT and HepG2-HIF1 α KO cells measured by HIF1 α immunofluorescence intensity (well-level average of per-cell mean intensity). One-way ANOVA followed by Tukey's post hoc test: ns: $P > 0.05$, **** $P < 0.0001$. **(b)** Comparison of phenotypic profiling between HepG2-WT and HepG2-HIF1 α KO cells by metabolism-focused biomarkers used in the manuscript.

3) The authors chose 1% oxygen for all their studies but if there is an adaptive response it would be expected to be graded along with oxygen availability. Are similar responses observed at other oxygen concentrations? For instance, is there a similar adaptive effect with mTOR/PI3Ki at <1% oxygen or 5% oxygen in the iPSCs or HepG2s?

In response to this request, we focused on 5% oxygen as a comparator to the reported 1% levels. (Unfortunately, our equipment is not able to reliably control oxygen levels <1%.)

We do observe a graded response (21%, 5%, 1% oxygen level) in HepG2 after 6d treatment. This is now reported for nuclear HIF1 α accumulation, inhibition of mTOR/BETs downstream targets (**Extended Data Fig. 6h**) and cell phenotypic response (**Extended Data Fig. 6i**).

Under the mild 5% hypoxia condition, cell survival was not affected in our functional assays for HepG2 (ischemia) and cardiomyocytes (hypoxia). Therefore, hit compounds were not further evaluated as there was no rescue to measure.

Reviewer 2:

The article by Li et al. presents a novel platform for high-content phenotypic screening to identify bioactive compounds that can accelerate cell adaptation to hypoxia, with the goal of alleviating acute hypoxic insult. Knowing that acute hypoxic stress can be detrimental to cells, leading to their irreversible damage and ultimately cell death, and that hypoxia preconditioning is one of the methods for cell adaptation to acute hypoxia-related stress, the authors hypothesized that shifting cells from a severe acute hypoxic state to a milder chronic state may provide a strategy to rescue cells from damage or death. The authors developed a microscopy-based, high-dimensional screening method designed to distinguish different cell states exposed to different oxygen conditions, ranging from normoxia (atmospheric oxygen levels) to acute hypoxia (1% oxygen for 6-48 hours) to chronic hypoxia (1% oxygen for 6 days). Using this platform and a large compound

library, they found that inhibition of mTOR or BET proteins in particular provides significant cytoprotection against ischemic stress.

Overall, the study is solid and interesting from both a methodological and biological point of view. In terms of methods, the authors relied on high-throughput fluorescence microscopy and bioinformatics to follow the changes in different cell phenotypes. Thus, this platform appears to be useful for applications aimed at investigating other stress-related conditions in cells. From a biological perspective, the study provides new insight into hypoxia-mediated changes in cells that could be modulated by inhibiting specific intracellular pathways. However, the authors used a cancer cell line to screen for stress-induced phenotypes. Since cancer cells differ from normal cells in their response to hypoxia, e.g. in terms of cell cycle or DNA damage, the presented results may not fully reflect the cellular milieu of normal cells. Thus, the compound screening data may be more appropriate for studies in cancer-related research, but not for any cell type. This should be better highlighted in the limitations of the study, as some phenotypes occurring in normal cells may not have been captured. In my opinion, the limitations are described too briefly. However, it must be acknowledged that the authors also used an iPS-derived model of oxidative damage in cardiomyocytes that showed beneficial effects of mTOR or BET inhibition, which is consistent with the data from HepG2 cells.

The analytical approach is sound and comprehensive. The methods are described in sufficient detail. Results are presented clearly and data analysis is performed accurately. There are only a few suggestions to consider:

1. The authors could elaborate a bit more on the advantages/disadvantages of the image-based phenotypic profiling presented in this study over other approaches to cell fate analysis, such as proteomic/kinetic/metabolomic. Could the authors add more on this point in the Introduction or Discussion?

We have added this point to the discussion. (lines 219 to 227).

2. The introduction is rather scarce. It could include more details on canonical and non-canonical mechanisms of hypoxia sensing in cells. This would help readers to better understand the background of the study.

We have added a summary of canonical and non-canonical mechanisms of cellular hypoxia sensing to the introduction part (lines 40 to 48).

3. If I am not mistaken, the code for comparing transcriptomic data for cancer cell lines and tissues is not provided.

We have added information to the Methods section to detail this analysis (lines 568 to 579). We also updated the zenodo link to include the script.

4. What post hoc test was used to calculate statistical significance after ANOVA? It is not specified in the text or figure legends (Figure 1, Figure 3, Extended Data Figure 3).

We used Tukey's correction for multiple comparisons as the post hoc test after ANOVA. This is now spelled out in the figure legends (lines 609, 611, 615, 639, 694) and the method section (line 324).

5. Why did the authors select 6 h of hypoxia treatment as the starting point for their analyses? It is known that hypoxia induces immediate changes in the activity of some proteins and in cellular metabolism. Please comment on this in your manuscript.

Thank you for this question, which was also raised by Reviewer #1, point 1. Our assay does detect phenotypic responses at earlier timepoints of 2h and 4h. Though, the differences among these early timepoints (2, 4, and 6hrs) are somewhat subtle. We have added these new data (Discussion lines 230-232, **Extended Data Figs. 6a, b**).

6. Extended data Figure 2 - the font size is too small. Please increase it, especially for the names of the color-coded pathways shown on the right.

We apologize for the small font size. This figure is now enlarged and provided as a separate figure (**Extended Data Fig. 3**).

7. "Fig. 2e" in the text (line 120) should be Fig. 2d.

This has been fixed. Thank you.

8. The authors state that: "Compounds associated with mTOR/PI3K and BETs were most effective in rescuing iPSC-CM's from hypoxic stress" [...] (lines 201-202), suggesting that other compounds were also tested. However, Figure 5C shows that only mTOR/PI3K and BETs inhibitors were used (although the legend in the scatter plot also indicates the use of other inhibitors in gray).

We apologize for the typo in the legend. The label 'Other' has been corrected to read 'DMSO'. So, all points are either mTOR/PI3Ki, BETi, or DMSO.

Reviewer 3:

The manuscript from Li et al describes the development of a high-content platform to screen for molecules that are able to facilitate adaptation to hypoxic stress. This approach was utilized to identify such compounds, which were found to largely involve mTOR/PI3K or BET inhibition activities, and were able to promote the survival of both liver cells exposed to ischemia-like stress and rescue cardiomyocytes from hypoxia stress. This work demonstrates a "ground-up" approach for assessing effectors of cellular phenotypes independent of target, and provides new means for identifying molecules with desired activities and new targets associated with biological processes, in this case hypoxia response and drivers to chronically adapted cellular states. The authors were thorough and did a nice job of presenting evidence to support their hypotheses. It is also appreciated that the compounds' effects were evaluated in two different cell types. The studies performed and results obtained are quite significant and merit publication in Nature Communications. There are some points that would be useful for the authors to address to improve the manuscript. Please find additional information in this regard in the comments below.

General/major comments:

1.) The authors should provide additional information regarding hypoxia adaptation responses in the introduction. It is only mentioned that there has been a focus on HIF/PHD/VHL in the introduction. At the very end (in the discussion), it is indicated that mTOR inhibitors and BET

inhibition have been used to alleviate hypoxia-related stress. As written, prior knowledge of the mTOR/PI3K/BET pathways' involvement and novelty of the findings presented are unclear.

We have added a summary of canonical and non-canonical mechanisms of cellular hypoxia sensing to the introduction part (lines 40 to 48), and a description of the current knowledge on mTOR/PI3K and BETs pathways in the context of hypoxia and hypoxic diseases in the Discussion (lines 246-257).

2.) The authors elected to use 6-day exposure to hypoxia as their model for chronic (adapted) hypoxia (and hence benchmark for compound evaluation). Was any longer period of time examined? It would be useful to know whether the 6-day mark was in fact adapted to the greatest extent possible or if it is only partially adapted. If in fact, longer than 6-days resulted in a "more adapted" state, would the number of hits (and their target distribution) change (i.e., which compounds would meet the new benchmark)?

The reviewer raised interesting questions about utilizing longer adaptation periods than 6d in our screen. We have now added characterization of HepG2 responses at 10d and 14d. As expected, cell state changed over time as assessed by HC phenotypic screening (**Extended Data Fig. 6c**). However, functional adaptation, as measured by ischemia tolerance, did not appear to improve after 6d (**Extended Data Fig. 6d**).

To address whether hits/targets would change, we reanalyzed our compound screen, this time using the newly established H10d or H14d point clouds (**Extended Data Fig. 6c**) as our "phenopushing" destinations. We found that 40% of the original hit compounds and original top enriched targets were consistent across all three analyses, with mTOR and BETs consistently among the most significant enriched targets for CH-phenopushing (**Extended Data Figs. 6e-g**). Interestingly, new hits/targets did arise, which focused on calcium signaling, PI3K-Akt pathway, and actin cytoskeleton pathways (enriched by new targets from H10d and H14d). While we originally chose 6d out of practical consideration, these suggested experiments certainly point out that there is more to discover.

We thank the reviewer for this suggestion, which is now presented in the Discussion (lines 228-234, 258-260 in Discussion).

3.) The authors should be clearer about the timing of compound treatment within the results section; the diagram in Figure 2a is a bit confusing. As written in the methods, it seems that the compounds are being used prior to initiation of hypoxia. It would be important to determine whether the molecules can initiate phenopushing in cells following the onset of hypoxia as well.

We apologize for the confusion. In our screen, cells were treated with compounds for 1d prior to initiation of hypoxia + 1d cotreatment with hypoxia (= 2d total time of drug treatment). We have revised the diagram (now **Extended Data Fig. 4a**) to clarify this.

To address the question of whether hit compounds can initiate phenopushing in cells following the onset of hypoxia, we considered 1d or 2d of total cotreatment time (with no pre-treatment), and we selected a subset of our hit compounds, primarily focused on the mTOR/BET targets. We found that 36/40 compounds (2d) or 26/40 (1d) triggered sufficient phenopushing to meet our hit cut-off (**Extended Data Fig. 6j**). These results show that our compound hits can initiate phenopushing even without pretreatment, and that longer treatment times provide stronger phenotypes.

We thank the reviewer for this suggestion, which is now presented in the Discussion (line 261-265).

4.) Having found that mTOR/PI3K/BET members were molecular targets of “hit” compounds, the authors assessed the down-regulation of these proteins’ activity across hypoxic states (Figures 3e-g). Can the authors provide some rationale for why deactivation might result in CH? This is especially interesting since pS6 and pAkt levels in CH are closer to those of AH, and in all three cases, normoxia has the highest levels, followed by AH and then CH.

We modified the writing in the results (lines 152-153, 159-161) to make the logic clearer. Namely, the activities of mTOR/PI3K/BETs monotonically decrease from N, to AH, to CH. The hit compounds, applied at AH, accelerate the decrease towards lower CH levels. We additionally added a discussion regarding mechanism as to how mTOR/PI3K/BETs deactivation leads to hypoxia adaptation (lines 247-253).

Specific/minor comments:

1.) It is very difficult to distinguish the circles used for 1 and 10 uM in Figure 4c. Can a mixture of colors and sizes (or some other approach) be used to make the figure clearer?

This has been fixed. Thank you.

2.) Lines 220-221 – “...to identify for compounds that push cell...” should be revised to read “...to identify compounds that push cells...”

This has been fixed. Thank you.

3.) In line 236, the authors discuss limitations of the study, which this reviewer does not believe is necessary. However, it would be useful if the authors utilized the discussion to present directions for further studies to study CH and these compounds.

We have added future directions into the discussion (lines 258-266).

4.) Line 323 – “...time...” should be “...times...”

This has been fixed. Thank you.

Point-to-point response.

Reviewer #1 (Remarks to the Author):

The authors have fully addressed all the concerns raised.

Thank you.

Reviewer #2 (Remarks to the Author):

I would like to thank the Authors for addressing all my concerns and recommendations. I have no further requests.

Thank you.

Reviewer #3 (Remarks to the Author):

This reviewer thanks the authors for their thoughtful responses and additional results presented in the present version of the manuscript. The only remark I have is the following:

The authors present new data in the current revision (Extended Data Fig 6 a-i), however their descriptions are limited to brief statements within the Discussion Section. It is recommended that these results be presented accordingly in the Results Section, with synopses/conclusions from them in the Discussion section.

Thank you for the suggestion. Fig S6 (the original Extended Data Fig 6) provides a coherent illustration into alternative strategies and future directions. As such, we feel Fig S6 is most logically placed within the narrative of the Discussion, where it will serve to inform and inspire future efforts.